# Modelling and Laboratory Tests of the Temperature Influence on the Efficiency of the Energy Harvesting System Based on MFC Piezoelectric Transducers

**DOI:** 10.3390/s19071558

**Published:** 2019-03-31

**Authors:** Marek Płaczek, Grzegorz Kokot

**Affiliations:** 1Institute of Engineering Processes Automation and Integrated Manufacturing Systems, Silesian University of Technology, Konarskiego 18a Street, 44-100 Gliwice, Poland; 2Institute of Mechanics and Computational Engineering, Silesian University of Technology, Konarskiego 18a Street, 44-100 Gliwice, Poland; grzegorz.kokot@polsl.pl

**Keywords:** Macro Fibre Composite, temperature influence, mathematical model, energy harvesting, energy efficiency analysis

## Abstract

Macro Fibre Composites (MFC) are very effective piezoelectric transducers that, among others, can be used as elements of energy harvesting systems. The possibility to generate electric energy, for example, from mechanical vibrations in order to power electrical elements that could not be powered in another way (using wires or batteries) is a great solution. However, such a kind of systems has to be designed by considering all phenomena that could occur during the exploitation of the system. One of those phenomena is the temperature fluctuation during the device operation. In the presented research work, a mathematical model of the energy harvesting system based on MFC transducers is proposed. The mathematical model was validated by laboratory tests conducted on a laboratory stand equipped with a universal mechanical testing machine (Instron Electropuls 10000) and a thermal chamber. During the tests, the samples were subjected to cyclic excitation simulating the operation of the system in various environmental conditions by forcing changes in the system operation temperature with the constant conditions of its excitation.

## 1. Introduction

The Macro Fibre Composite (MFC) is a leading low-profile actuator and a sensor offering high performance, flexibility and reliability in a cost-competitive device [1]. Composite piezoelectric materials have been developed as a solution to eliminate the limitations in practical applications of classic, monolithic piezoelectric transducers. Their basic disadvantage is fragility, which can lead to damage during handling and assembly, or the difficulty of their application on curved surfaces. The limitation of applications, especially in the case of light constructions, is also their high density and the resulting large mass of classic ceramic piezoelectric transducers. Thus, composite piezoelectric materials in the form of piezoceramic fibres embedded in a polymer matrix have been developed. Such materials are the Active Fibre Composite (AFC) and Macro Fibre Composite (MFC). Developed non-classical piezoelectric transducers could be applied in a wide range of systems, for example, in passive or active vibration control, energy harvesting, structural health monitoring or morphing structures [2,3,4,5,6,7,8,9].

The possibility of applying composite piezoelectric transducers in systems recovering electric energy from mechanical vibrations is very interesting for researchers. So far, the amount of obtained electric energy has been not considerable but sometimes it is enough to power some elements like microcontrollers or other electronic devices with low energy consumption [10,11,12]. The main advantage is that there is no necessity to replace batteries or use wires. 

This is why the research on harvesting energy from other alternative sources has quickly become a new area that could lead to the avoidance of problems with powering systems with very low power needs, located in places where the traditional power sources cannot be used. For example, in Reference [13], the authors have undertaken to investigate the possibility of acquiring electricity from the movement of muscles inside the body to power pacemakers. The PZT material was placed on a polyamide tape, which after previous bio-neutrality tests with smooth muscle cells of mice, showed no negative effect on cell reproduction and metabolism. Tests carried out on beef and sheep hearts showed that the efficiency of the piezoelectric material differs depending on the place of its attachment (right ventricle, left ventricle, free wall), as well as on the angle of its attachment in a given area of the heart. Depending on the organism, the tapes in the most optimal place were able to generate up to 4 V on the beef heart and 2 V on the sheep one. The diaphragm was the next muscle on which the PZT MEH tape was placed. Another example is the generation of electric energy from walking [10]. For the construction of footwear with piezoelectric strips, tapes developed by NASA and produced by Face International Corp. were used. The system bent in a way that increases its effectiveness system, placed in sports footwear between the sole and the insole, exactly under the heel. Such shoes, when moving at a speed of one step per second, could generate about 100 mW of electric energy. By loading each strip with a 250,000 ohms resistor, the PZT belt generated up to 150 V. The very interesting conception of the double pendulum based piezoelectric energy harvester system in an attempt to maximise the energy harvested from common human motions is proposed in Reference [7]. This setup consists of a piezoelectric cantilever beam with a magnet as the tip mass, attached to a double pendulum. A great increase in the maximum output voltage and power was achieved with the proposed system on both the well-controlled mechanical shaker test and practical tests on human arm and leg motions in the walking and jogging modes. Additionally, in Reference [8], a novel energy harvester which concurrently harnesses energy from base vibrations and wind flows is proposed by utilizing a mechanical stopper to achieve a broadened bandwidth. The authors proposed an aero-electro-mechanically coupled model by considering the mutual coupling behaviours between the harvester structure, piezoelectric transducer and airflow, and the impacting behaviours between the harvester and the mechanical stopper. Tests conducted by the authors showed that at a wind speed of 5.5 m/s and a base acceleration of 0.5 g, the output power from the proposed harvesting device steadily increases from 3.0 mW at 17.3 Hz to 3.8 mW at 19.1 Hz. Electric energy could be also collected from falling water drops. As it is presented in Reference [14], the energy of a drop of water falling on a dry surface of the system is not fully absorbed, but a part of it is used to spray drops of water, decompose and partially recoil. With the increase in the amount of water on the surface of the piezoelectric system, it was noted that with each next drop, more and more energy is absorbed, which translates into the efficiency of the entire system. Therefore, the energy recovered from one drop falling on a dry surface is much smaller than the energy of a drop of water falling on a previously formed water lens on the surface of the piezoelectric system. Another example is the use of airflow for generating electric energy. Harvesting energy from Von Karman whirls can be very effective, as it is shown in Reference [15]. In Reference [16], the authors showed that the Macro Fibre Composite is well suited to harvest energy directly from waste vibration present in the environment without additional mechanical fixtures due to its flexibility, robustness and long lifetime. They claimed that it overcomes the disadvantages of a monolithic piezoceramic energy harvester requiring additional mechanical fixtures to transfer vibrations in a resonant and controlled setup. This allows for completely new and cost-effective applications of harvesting energy from structural vibration sources. However, they pointed out that to design energy harvesters, a specific and careful adaptation to each targeted vibration source for maximum efficacy is required. The energy harvester can compete with batteries in terms of overall costs in low power applications if these applications require a long lifetime and/or with difficult or impossible maintenance access, an increased operational temperature range. As pointed out, they can work in a wide temperature range, i.e., from −20 °C up to +100 °C [16].

Taking into account the examples presented above which are only a small part of the literature on the subject, it can be clearly noticed that a lot of phenomena can be used to generate electric energy using piezoelectric transducers that generate it from mechanical vibrations [7,8,9,10,13,14,15,16]. However, there are some external independent parameters (environmental parameters) that could have a strong influence on the energy harvesting system efficiency. Such a parameter could be an operation temperature which is considered, for example, in Reference [17,18,19,20,21,22,23,24,25,26].

In Reference [17], three kinds of ceramic–polymer composite piezoelectric materials were evaluated and compared against state-of-the-art piezoelectric materials. The new composites are unstructured and structured composites containing granular lead zirconate titanate (PZT) particles or PZT fibres in a polyurethane matrix. The composites were used to build energy harvesting patches which were attached to a tyre and tested under simulated rolling conditions. As the authors mentioned, the energy density of the piezoelectric ceramic–polymer composite materials was initially not as high as that of the reference materials: MFC transducers and polyvinylidene fluoride polymers. However, the area normalized the power output of the composites after the temperature and strain cycling was comparable to that of the reference devices because the piezoelectric ceramic–polymer composites did not degrade during the operation. The authors of this work also analysed the decrease in room temperature constants due to thermal degradation at elevated temperatures. They informed that the changes in a piezoelectric constant measured in the sample were small and could be the result of the high-temperature relaxations in the material while the PVDF piezoelectric material had a poor temperature stability and the degradation of the MFC above 90° C was also significant.

In Reference [18] the author mentioned that the unbalanced bridge constitutes a very important issue when working with a self-sensing circuit. The self-sensing actuator is described as the system that works through circuitry that can distinguish between the sensing signal and the control signal applied to the piezoelectric patch. In doing this, the circuit can cancel the applied control voltage out and return only the sensing signal. It allows one piezoelectric element to sense and apply actuation to the beam at the same time. The unbalanced bridge in this system means that the value of the piezoelectric capacitance and the matched capacitance are not exactly the same. This can occur for a number of reasons including accidental mismatch, piezoelectric ageing, and ambient temperature changes. According to the author, the piezoelectric material has a capacitance that depends inter alia on temperature and because of that, the circuit can become unbalanced. This phenomenon, as well as others, have limited the use of the self-sensing actuators in many practical applications.

Reference [19] presents the design, fabrication and testing of a Macro Fibre Composite (MFC) based bimorph actuator to meet stringent requirements such as size, stroke and thermal stability. The authors aimed to develop a small (25 × 5 mm) actuator that can generate a large stroke (1 mm) relative to its size and is stable over a large temperature range (from −30 to 70° C) with a small variation (<0.1 mm) in the actuator position over that temperature range and engagement within 10 ms. They created the bimorph using MFC transducers with an average value of the thermal deflection staying well within the specified limits. It was evaluated in terms of use in space-constrained applications. The actuator fabrication was followed by its characterization at room temperature and at an elevated temperature. The authors claimed that on the basis of an overview of the available actuation technology and shape changing materials, Macro Fibre Composite (MFC) actuators could be identified as suitable for this application.

In Reference [20], an experimental study of the effect of temperature on the electrical impedance of the piezoelectric sensors used in the EMI (electromechanical impedance) technique for structural health monitoring is presented. The authors used 5H PZT (lead zirconate titanate) ceramic sensors in their research. The authors pointed out that temperature effects have been cited in the literature as one of the most critical and challenging of several practical problems. The experimental results presented in the paper showed that the temperature effects were strongly frequency dependent. The authors investigated the effect of temperature on the electrical impedance signatures of a conventional 5H PZT sensor used in structural health monitoring. The variations in both the amplitude and the frequency were analyzed experimentally by using an aluminium specimen and obtaining impedance signatures at temperatures ranging from 25 °C to 102 °C. The experimental results showed that the variations in the amplitude of the impedance signatures were related to the temperature-dependence of the capacitance of the piezoelectric sensor. 

The temperature effect in structural health monitoring is described also in Reference [21]. Temperature effects in the ultrasonic Lamb wave structural health monitoring systems are analysed. A model is proposed to account for all relevant temperature-dependent parameters of a pitch-catch system on an isotropic plate which was used to predict the response spectra in aluminium plates for the temperature range of −40 to +60 °C, which accounts for normal aircraft operations. It was found that the changing temperature has a negligible effect on the wavelength tuning points and a marked effect on the response amplitude. The results presented in the paper apply to PZT-5A and MFC type P1 transducers bonded on aluminium plates with ambient-cure epoxy. The authors suggest that the presented study can be used to develop strategies for the compensation of the temperature effects in the guided-wave damage detection systems. 

Moreover, in Reference [22], the influence of temperature as well as bonding thickness on the actuation of a cantilever beam by PZT patches was analysed. The authors wanted to investigate the sensitivity of transferring the piezoelectric actuation efficiency to a cantilever beam due to variations in the ambient temperature and possible variations in the thickness of the adhesive layers bonding actuators to a beam. They described a static, analytical model with a linear longitudinal strain distribution across the whole actuated zone of a PZT actuated cantilever beam including the bonding layers. As they proved, the reduction in the actuation induced longitudinal strain grows relatively strongly as a function of the adhesive layer thickness, but it is quite negligibly affected by a change in the temperature dependent adhesive material. They claimed that together with another shear transfer model, this analysis of the piezoelectric actuation in the presence of bonding layers confirms, however, that the actuation efficiency transfer cannot be altered enough in normal working conditions. The authors presented results of dynamic FEM calculations of the piezoelectric–mechanical fields done with overall temperatures varying from −20 to 50 °C and also with thicker adhesive bonding layers up to 200 μm. The presented results show only a small shift in the resonance frequencies without any change in the amplitude of the acceleration peaks. The FEM analysis of the temperature influence has been validated experimentally.

In Reference [23], the actuation performances of piezoelectric fibre composites (PFCs) were studied when they were exposed to ambient temperatures from −15 °C to 80 °C. PFCs are often used in various aerospace structures and they are exposed to a wide range of temperatures during exploitation while the temperature effects on the mechanical properties of PFCs are important. The results presented by authors showed that the actuation performance of PFCs greatly depended on the ambient temperature. They inform that the free strain values and the calculated piezoelectric coefficients d33 and d31 were initially enhanced with the increase of temperature due to the restricted movement of the epoxy molecule in the glassy state. The actuation performance decreased as the temperature increased above room temperature since the deformation of epoxy molecules in the viscoelastic state was irreversible.

In Reference [24], the constant-field (short circuit) thermoelastic properties while developing equations for the coefficients of thermal expansion (CTE) of the MFC as a function of temperature were considered. The temperature range analyzed by the authors was from 0 to 250 °C and the analytical results were verified using the ANSYS finite element software. The authors presented the coefficients of thermal expansion for the MFC actuator as a temperature function, based on finite element and classical lamination analyses. It was found out that while the coefficient of thermal expansion in the direction along the length of the MFC is nearly constant between 0 ° and 250 °C, the coefficient of thermal expansion in the width direction varies significantly, primarily due to the thermoelastic behaviour of the epoxy matrix. As a result, the variation of thermoelastic properties with temperature should be carefully modelled, particularly when the transverse behaviour of the MFC is deemed critical.

In Reference [25], the authors analysed the effect of temperature changes on the piezoelectric parameters of a soft PZT Ceramic. They investigated the temperature behaviour of the main piezoelectric parameters, such as the electromechanical coupling factor, the mechanical quality factor, the dielectric permittivity, the loss tangent, the piezoelectric charge constants *d*_33_ and *d*_31_ and the voltage constants *g*_33_ and *g*_31_ of a soft type piezoelectric material. They showed that temperatures up to 150 °C do not influence the piezoelectric parameters. The authors indicate that every transducer made from this type of material may be successfully used up to 150 °C. The authors showed that between this temperature and 250 °C, the piezoelectric properties undergo more or less important changes, mainly due to the depoling effect and above that, at temperatures over 250 °C, they degrade very rapidly, tending to zero. In the paper, the variation of the main piezoelectric parameters of a soft type PZT material was studied as a function of temperature on a large temperature interval, from room temperature up to the Curie point. The authors did not study the behaviour of the tested piezoelectric material at temperatures below zero Celsius degrees. The considered piezoelectric material was prepared with the usual ceramic method. The sintered samples were mechanically processed as discs with plane parallel faces having a diameter of 10 mm and the thickness of 1 mm and as parallelepipeds of 20 × 5 × 5 mm using special cutting and polishing machines. As the authors informed, these dimensions are standard ones, required to determine correctly the material parameters. 

In Reference [26], the authors pointed out that the piezoelectric material usually works at various temperatures but the characterization of the material is mostly performed and reported in the literature at room temperature which leads to incorrect analysis results and the inability to properly design the system and optimize its operation under given conditions. They informed that the depolarization in piezoelectric material occurs when the material is heated to its Curie temperature and when the mechanical stresses are high enough to disturb the properties of the material. They studied the performance of the lead zirconate titanate (PZT-5A) piezoelectric material under various temperatures and loading conditions and showed that the output voltage of the piezoelectric material decreases with the increase of temperature. What is more, they found out that the output voltage from the harvester increases when loading increases while its temperature decreases. In their research work, a rectangular PZT-5A specimen with dimensions 20 × 10 × 5 mm was selected for experimentation. They observed that at the temperature of 20 °C different loading conditions do not influence the output voltage so a transducer made from this type of material may be successfully used up to this temperature. When the temperature was between 50 °C and 300 °C the properties of piezoelectric ring change due to the depolarization effect and the output voltage decreases. Of course, at the temperature of 350 °C, they degrade very rapidly and tend to zero. 

Following the literature review, MFC piezoelectric transducers can be successfully used in a variety of applications. However, one should remember the influence of temperature on the efficiency of their operation. As the authors of the studies [17,20,21,23,26] pointed out, changes in the operating temperature of the system already in the range of relatively small deviations from room temperature can significantly influence its parameters and efficiency. On the other hand, in References [24,25], the authors showed the stability of classic piezoelectric transducers PZT up to a temperature of 150 °C, as well as small values of thermal expansion of composite transducers. Therefore, the authors of this study decided to undertake the analysis of the topic and carry out experimental research, as well as to propose a mathematical model of the system in which the influence of temperature on its efficiency is taken into account.

## 2. Materials and Methods

The paper presents a mathematical model of the system harvesting electric energy from mechanical vibrations using the MFC piezoelectric transducer. Using the constitutive equations and the analysis of alternating current circuits, the value of the voltage drop on the resistor connected to the transducer terminals was determined. Then, laboratory tests of the analysed system were carried out in order to verify the proposed mathematical model of the energy recovery system. In addition, a series of laboratory tests of the system operation in the temperature ranging from −30 to +70 °C were carried out to determine the effect of temperature on the efficiency of converting mechanical energy into electrical energy through the used piezoelectric transducer.

Piezoelectric materials can be described by means of a pair of constitutive equations, in which relations between mechanical and electrical properties of transducers are included [2,27,28]. Taking into account the method of loading a piezoelectric transducer, in the case under consideration, these equations can be written as
(1)$$D_{3}=\varepsilon_{33}^{T}E_{3}+d_{31}T_{1},$$
(2)$$S_{1}=d_{31}E_{3}+s_{11}^{E}T_{1},$$
where $\varepsilon_{33}^{T}$, $d_{31}$
$s_{11}^{E}$, are the dielectric, piezoelectric and elastic compliance constants. The upper indexes *T* and *E* denote, respectively, the fixed value at constant/zero stress or constant/zero value of the electric field. In the case under consideration, the $d_{31}$ piezoelectric constant was included in the constitutive equations. The subscripts indicate that it is a constant, which describes the relationship between the transducer stress in the direction of axis 1 and the electric field in the direction of axis 3 (the transducer electrodes are perpendicular to axis 3). The determinations were made in accordance with the standard given by IEEE [2,27,28]. Symbols *D*_3_, *S*_1_, *T*_1_ and *E*_3_ denote the electric displacement, strain, stress and intensity of the electric field in the directions of the axes described by the lower indices of symbols. At this step, in the presented form of the constitutive equations, there are no components introducing the temperature influence. In the proposed mathematical model, the influence of temperature changes on the efficiency of the system’s operation will be introduced by taking into account the temperature dependence of the relative dielectric constant. Equations taking into account these relationships have been developed in Reference [27] for crystals with piezoelectric properties. These compounds could be included in the proposed process of modelling composite piezoelectric transducers which will be the subject of further research conducted by the authors and will be presented in subsequent publications. The piezoelectric transducer equation, with an attached resistor, was derived using the classical method of transient analysis of linear electrical circuits. The piezoelectric transducer can be modelled as a source of electric charges with a defined electrical capacity. The piezoelectric transducer with the external electrical circuit in the form of a resistor connected to its terminals was modelled as an RC type system in an alternating current circuit, the source of which is a piezoelectric transducer [28]. Assuming that the initial state in the considered circuit is zero, the capacitor is not pre-charged after the harmonic voltage $U_{P}\left(t\right)$ generated by the piezoelectric transducer is fed into the circuit as a result of its deformation, a transient state occurs in the circuit. 

Taking into account the electric substitute diagram of the piezoelectric transducer described in details in other work [28] and the equations known from the analysis of alternating current circuits (the RC electric circuit in this case), according to the second Kirchhoff’s law, the following equation can be derived:(3)$$R_{Z}C_{P}\frac{\partial U_{C}\left(t\right)}{\partial t}+U_{C}\left(t\right)=U_{P}\left(t\right),$$
where *R_Z_* means the resistance of the connected resistor, *C_P_* the electric capacity, and $U_{C}\left(t\right)$ the voltage on the covers of the capacitor. The voltage generated by the piezoelectric transducer, under the influence of deformations caused by the external force is
(4)$$U_{P}\left(t\right)=\frac{Q\left(t\right)}{C_{P}},$$
where Q(t) means the generated electric charge. By determining the stress value of the piezoelectric transducer from constitutive Equation (2) and substituting the obtained dependence to Equation (1), after the ordering, the following was obtained:(5)$$D_{3}\left(t\right)=\frac{d_{31}}{s_{11}^{E}}S_{1}\left(t\right)+\varepsilon_{33}^{T}\left(1-\frac{d_{31}{}^{2}}{s_{31}^{E}\varepsilon_{33}^{T}}\right)E_{3}.$$

Equation (5) takes into account the variation in time of electrical displacement resulting from the time-varying transducer deformation. The electrical displacement is defined as the electric charge per unit area. The surface of the electrodes in the MFC transducer can be determined as the surface area of the active part of the transducer multiplied by a copper fibre volume fraction $V_{E}=0.190$ [24]. Taking into account the dependence of the intensity of the electric field defined as the ratio of the voltage on the transducer electrodes to its thickness, and taking into account that in the considered case the electric charge collected on the surface of the electrodes is a function of time, the voltage generated by the piezoelectric transducer can be described by
(6)$$U_{P}\left(t\right)=AS_{1}\left(t\right)+BU_{C}\left(t\right),$$
where,
(7)$$A=\frac{V_{E}l_{P}b_{P}d_{31}}{C_{P}s_{11}^{E}},$$
(8)$$B=\frac{V_{E}l_{P}b_{P}\varepsilon_{33}^{T}}{C_{P}h_{P}}\left(1-k_{31}{}^{2}\right),$$
(9)$$k_{31}{}^{2}=\frac{d_{31}{}^{2}}{s_{11}^{E}\varepsilon_{33}^{T}}.$$

The symbol $k_{31}^{2}$ indicates the electromechanical coupling constant, by means of which the efficiency of converting mechanical energy into electric energy as well as electric energy into mechanical energy by the piezoelectric transducer is determined [2]. Equation (6) describes the voltage generated by the piezoelectric transducer, assuming its uniaxial, homogeneous deformation. Taking into account that in the case of the RC circuit:(10)$$U_{P}\left(t\right)=U_{C}\left(t\right)+U_{R}\left(t\right),$$
where $U_{R}\left(t\right)$ means the voltage drop across the resistor connected to the MFC transducer. Taking into account that in the case of the RC circuit under analysis the total resistance of the system is determined by the value of the applied resistor *R_Z_*, the voltage drop measured on the resistor is
(11)$$U_{R}\left(t\right)=\frac{U_{P}\left(t\right)}{\sqrt{R_{Z}{}^{2}+{\left(\frac{1}{\omega C_{P}}\right)}^{2}}}R_{Z}.$$

By substituting Equation (6) to Equation (11), the voltage drop at the resistor connected to the transducer can be described as the function of deformation of the transducer with the following dependency:(12)$$U_{R}\left(t\right)=\frac{AR_{Z}}{BR_{Z}+\left(1-B\right)\sqrt{R_{Z}{}^{2}+{\left(\frac{1}{\omega C_{P}}\right)}^{2}}}S_{1}\left(t\right).$$

Assuming the parameters of the analysed system, which are summarized in Table 1, the peak value of the voltage drop on the connected resistor was determined. 

The obtained results were compared with the value of the voltage drop on the resistor connected to the MFC transducer measured during the experimental tests.

In order to properly clamp the tested transducer into the testing machines, jaw mounting elements were designed and prepared using a 3D printer, made of High-Performance HTPLA-CF material [29]. This durable PLA has a Heat Deflection Temperature (HDT) of more than 140 °C after heat treating. This material was used because the parts made of it maintain strength and form up to much higher temperatures than PLA, ABS, or Polyesters which was very important in the conducted tests. The assembly elements were designed in such a way as to enable the safe sticking of the examined piezoelectric foil and its correct operation during the tests. In the assembly elements, a cavity of a thickness corresponding to the thickness of the MFC transducer was made for this purpose in order to protect it against crushing during the assembly in the jaws of the machine. The piezoelectric transducer was pasted using a cyanoacrylate adhesive. Due to the method of fixing the film in the jaws of the machine, the active length of the piezoelectric transducer depicted in Table 1 was limited to a length of $l_{MFC}=52\;\left(mm\right)$. The designed mounting elements and a ready to use sample of the MFC piezoelectric transducer are presented in Figure 1.

The terminals of the tested piezoelectric transducer were connected to an external resistor (placed outside of the thermal chamber) and an electric voltage drop was recorded on it using a National Instruments measuring card NI9215 [30] with the measuring range ±10 V. The measurement card was installed in the cDAQ-9191 CompactDAQ Chassis [31] and connected to a computer, by means of which the measurement data was recorded using LabVIEW software. The sampling frequency during measurements was set to 100 S/s. Data recording took place during the tests after starting the recording by a voltage trigger issued by the analogue output of the Instron controller only during the excitation of the test samples. The view of the test sample mounted in the jaws of the testing machine generating excitation of the sample is shown in Figure 2a and the extensometer installed on the tested sample is presented in Figure 2b.

The tests of the MFC piezoelectric transducer were performed using the universal electromechanical testing machines Instron ElectroPuls E10000 (Instron, HighWycombe, UK), equipped with a temperature chamber and an extensometer. The tests were performed under strain control. The test method was as follows: in each stage, the temperature was raised up by 10 °C (start point *T* = 20 °C), and then the temperature stabilization step began. After a period of temperature stabilization (time equal $t_{s}$ = 10 min) the tested piezoelectric transducer was stretched up to the mean value of strain ε_M_ = 0.03 and then subjected to fluctuating cyclic strain load with the amplitude $\varepsilon_{A}$ = 0.02, frequency *f* = 1.5 Hz, no. of cycles *n* = 500. For technical reasons, the research was divided into two stages. The procedure was applied starting from a temperature point $T_{S}$ = 20 °C with a temperature increment equal to $T_{i}$ = 10 °C up to the end point $T_{E}$ = 70 °C, and the second test was within the starting point temperature $T_{S}$ = 20 °C with an increment equal to $T_{i}$ = −10 °C up to $T_{E}$ = −30 °C. The useful working range given by the manufacturer is from −35 to +85 Celsius degrees [1]. The Instron controller analogue output voltage signal was streaming to the National Instrument controller using the analogue output as the trigger signal for data acquisition (changes in generated piezoelectric transducer electric voltage). The laboratory stand was controlled using an extensometer, ensuring a constant deformation value of the transducer at the maximum level of 2000 ppm. 

The conducted tests were limited to the constant value of the excitation frequency despite the fact that in Reference [20], the experimental results showed that the temperature effects were strongly frequency-dependent. At this stage of the research, however, it was assumed that the modelled system will be subjected to constant-frequency forcing, as in the case of the solution presented in Reference [10], where the system was designed to recover energy from human walking and running. Therefore, the frequency of extortion was assumed to be consistent with the frequency of the impact of the foot against the ground during the run and it was assumed as equal to 1.5 Hz. The results presented in Reference [20] have become an incentive for further studies of the temperature-frequency relationships of the system.

## 3. Results

On the basis of the mathematical model of the investigated system of energy harvesting from mechanical vibrations using the MFC piezoelectric transducer, the peak value and the time course of the voltage drop on the resistor connected to the transducer terminals were determined using Equation (12). Figure 3 presents the analytically determined course of the voltage drop across the resistor juxtaposed with the values recorded during measurements at the laboratory stand carried out at 20 Celsius degrees. The parameters of the MFC M8514P2 transducer set out in Table 1 were taken into account for calculations including the limited value of the active length resulted from the mounting system of the sample. The resistance of the transducer connected to the piezoelectric transducer as well as the transducer excitation parameters are listed in Table 2.

A high consistency of results obtained in laboratory conditions with calculations using the proposed mathematical model was shown. The voltage peak values are respectively 3972 V for the analytically determined voltage and 4397 V for the voltage drop measured on the resistor during laboratory tests (the average value of the peak voltage value from 500 load cycles of the tested sample was determined).

In the next stage of the research, an experimental analysis of the impact of changes in the working temperature of the tested system on the efficiency of electricity generated by the used MFC piezoelectric transducer was done. The obtained results in the form of peak values of the voltage drop measured on the resistor connected to the transducer are presented in Figure 4. By analysing the obtained waveforms of the measured voltage drops on the resistor at individual system operating temperatures by the FFT analysis, a fixed frequency of 1.5 Hz was confirmed. By providing a constant value of deformation of the tested sample by using an extensometer and providing a constant frequency, identical excitation parameters of the tested sample were ensured during all research tests.

Analysing the obtained results of laboratory tests, a significant effect of temperature change on the voltage drop value on the resistor connected to the tested MFC piezoelectric transducer can be observed. There is a decrease in the voltage peak value by more than 50% along with the temperature increase in the analysed range from −30 to +70 degrees Celsius. Such an important change translates directly into the efficiency of the system harvesting energy from mechanical vibrations exposed to variable temperature operating conditions. One of the reasons for the decrease in the efficiency of the system as the temperature rises may be the change in the dielectric constant value of the material with piezoelectric properties, which is applied to the tested composite transducer.

In order to carry out the analysis and verification of the mathematical model developed for the considered energy recovery system, the dependence of the dielectric constant on the temperature was taken into account. The course of the dependence of the dielectric constant of PZT transducers on temperature can be found, among others, in References [32,33,34]. In Reference [34], a graph of the temperature dependence of the relative dielectric constant of the PZT transducer is presented. This paper proposes a method for the composition and synthesis of lead zirconate titanate (PZT) piezoelectric ceramic for use in energy harvesting systems. The proposed material consists of $\left(1-x\right)Pb\left(Zr_{0.53}Ti_{0.47}\right)O_{3}-xBiYO_{3}$−x) [PZT–BY(x)] (x = 0, 0.01, 0.02, 0.03, 0.04, and 0.05 mol) ceramics near the morphotropic phase boundary (MPB) region, prepared by a solid-state mixed-oxide method. As the authors describe in the paper, the relative dielectric constant has a small linear increase up to 300 Celsius degrees, and then increased very fast to a maximum value at the Curie temperature. Thereafter, the relative dielectric constant decreases with further increases in temperature. This dielectric anomaly indicates a phase transition from the ferroelectric to paraelectric phase at the Curie temperature. The Curie temperature of pure PZT is about 360 °C. Upon adding 0.01 mol of BY, the Curie temperature shifted towards a higher temperature (about 373 °C) [34]. Very similar results are also presented in References [29,30]. 

Therefore, a linear change in the dielectric constant value can be assumed in the tested range of the system operating temperature. A linear change of values of the dielectric constant in the studied temperature range was assumed from $\varepsilon_{33}^{T}=0.435E-08\;\left[\frac{F}{m}\right]$ at −35 Celsius degrees up to $\varepsilon_{33}^{T}=2.575E-08\;\left[\frac{F}{m}\right]$ at +70 Celsius degrees. The value presented in Table 1 was assumed for the room temperature (20 °C). Peak values of the voltage drop on the resistor attached to the tested piezoelectric transducer were determined analytically using the derived dependence (2.12). The graph of changes in the peak voltage drop on the connected resistor is shown in Figure 5. The obtained values were compared with the values measured during experimental studies.

## 4. Discussion

Taking into account the literature review as well as the presented results, it is very important to consider the influence of the temperature change on the efficiency of the energy harvesting systems. For the considered Macro Fibre Composite piezoelectric transducers the useful working range given by the manufacturer is from −35 to +85 Celsius degrees. The most important temperature limits are the Curie temperature of the PZT fibres and the glass transition temperature of the epoxy that is used inside the MFC. A good rule is that a piezoelectric transducer should not exceed half of the Curie temperature during operation. In the case of MFCs, it is about 150 Celsius degrees. However, the load of the transducer should be also considered when setting working temperature limits. The designer should take into account that the temperature thermoset material gets compliant or viscoelastic so that probably the most of the generated strain gets lost by the shear deformation of the epoxy matrix in this case [1]. 

The tests conducted in laboratory conditions proved that temperature changes could significantly affect the efficiency of the energy harvesting system based on Macro Fibre Composite applications. By providing a constant value of deformation of the tested sample by using an extensometer to control the movement of the machine jaws and providing a constant frequency, identical excitation parameters of the tested sample were ensured during all research tests. Only temperature changes could have an influence on the obtained results. The laboratory stand and tested pieces were prepared in such a way to omit the influence of using different epoxides and substrate materials during the testing influence of the temperature. This is why the tested MFC was not attached to any kind of vibrating mechanical subsystem using a glue layer but the transducer itself was stretched directly. This allows the omission of the influence of other elements on the efficiency of the energy harvesting system based on using the MFC transducer. However, in real conditions, these elements are often glued on the surface of a vibrating mechanical subsystem and the influence of the connecting layer should also be taken into account. Issues concerning the influence of the adhesive layer are also the subject of research both by the authors of this work [28] and by other researchers [22].

The loss obtained at the measured voltage drop on the externally applied resistor could partially result from the dielectric constant changes caused by the different temperature of the sample operation. The dependence of the PZT piezoelectric material’s dielectric constant value on the temperature was shown also by other researchers inter alia in References [32,33,34]. However, those changes are rather small in the range of analysed temperatures of the system operation. It can be also assumed that changes in the dielectric constant value are linear in the analysed temperature range. By juxtaposing the results of the laboratory tests with the results of analytical calculations based on the proposed mathematical algorithm the compliance of the trend of the decrease in the value of the voltage generated together with the increase of the operating temperature of the tested system could be shown. 

However, the consistency of the results could be raised by developing a mathematical model of the system and a more detailed description of the phenomena occurring in the system under study. To improve the mathematical model of the system, the relationship between the elastic, dielectric and thermal properties of piezoelectric materials should be taken into thought. The equations taking into account these relationships have been developed in Reference [27] for crystals with piezoelectric properties. They were obtained from the solution of thermodynamic compounds in the crystal. The mathematical analysis of thermodynamic equations allowed the authors of the study to define individual material constants and to determine the relationships between them, also taking into account the thermal properties of crystals. The obtained equations describing the elastic, dielectric and thermal properties of crystals form a system of equations that fully characterizes the properties of piezoelectric crystals. These compounds can be included in the process of modelling composite piezoelectric transducers based on the use of piezoelectric fibres, which is presented in this study. Additionally, the influence of the viscoelastic phenomenon of the epoxy resin constituting the matrix of piezoelectric fibres can be included in the mathematical model of the system. This will lead to the better mapping of the phenomena occurring in the tested transducers with a simultaneous significant impact on the complexity of the mathematical description. These issues will be the subject of further research conducted by the authors and will be presented in subsequent publications. The mathematical model presented in this work allows for the correct estimation of the efficiency of the modelled system of energy harvesting from mechanical vibrations, taking into account the temperature of the system operation. Moreover, the proposed model is not complex and is easy to use with regard to the modelling of many applications of MFC piezoelectric transducers.

## 5. Conclusions

A significant effect of temperature changes on the efficiency of the system designed for energy harvesting from mechanical vibrations based on the use of the Macro Fibre Composite piezoelectric transducer was presented. Both the modelling process of the analysed system and analytical determination, as well as experimental determination during laboratory tests of the electric voltage on the resistor connected to the MFC transducer terminals, were presented. The laboratory stand and tested pieces were prepared in a way to omit the influence of other elements during testing the influence of the temperature in order to clearly verify its impact on the system efficiency. The tested MFC transducer was subjected to excitation in the form of an axial force causing harmonic deformation with a constant amplitude and frequency. Voltage measurements were carried out in the range from −30 to +70 Celsius degrees. As it was shown, the decrease in the efficiency of the system with the increase in its operating temperature is, among others, caused by the change in the dielectric constant of the piezoelectric material from which the transducer’s fibres were made. Compliance of the results of mathematical modelling with the measurements carried out on the real object has been shown. However, the possibility of further development of the proposed mathematical apparatus was indicated for the inclusion of the relationship between the elastic, dielectric and thermal properties of piezoelectric materials; the influence of the viscoelastic phenomena of the epoxy resin constituting the matrix of piezoelectric fibres as well as the temperature-frequency relationships of the system. Issues concerning the influence of the adhesive layer will also be the subject of future research.

## Figures and Tables

**Figure 1 sensors-19-01558-f001:**
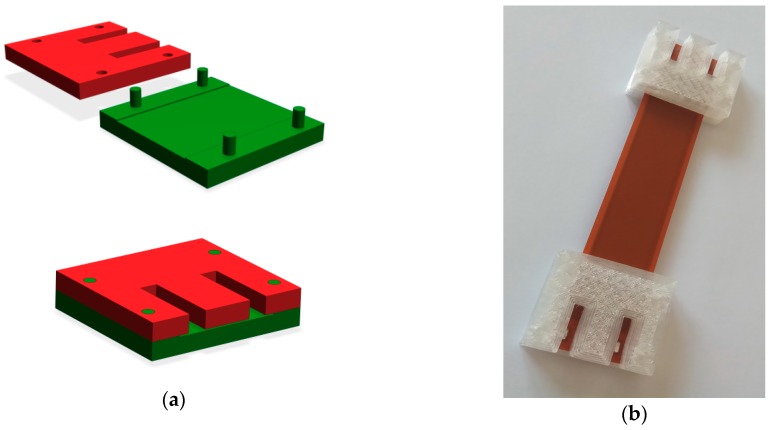
The mounting elements (**a**) and a sample prepared for testing (**b**).

**Figure 2 sensors-19-01558-f002:**
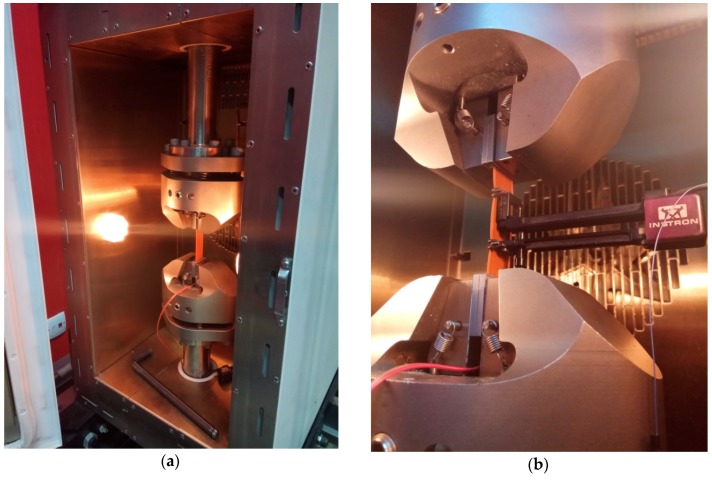
The sample fastened in the jaws (**a**) and an extensometer installed on the tested sample (**b**).

**Figure 3 sensors-19-01558-f003:**
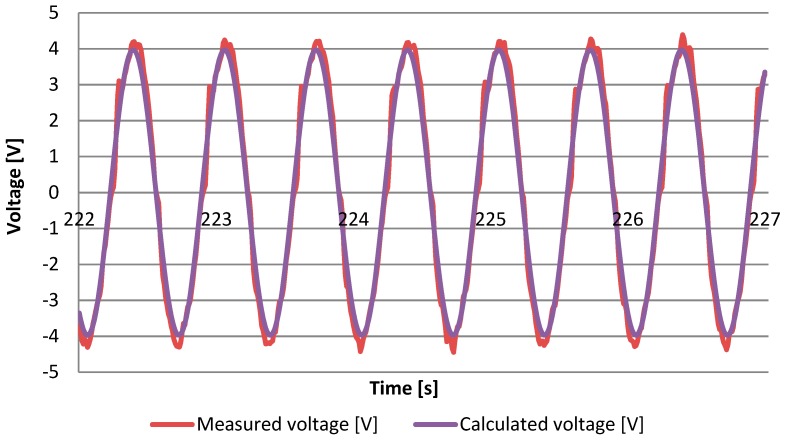
The analytically calculated voltage drop on the resistor and the course recorded during measurements at the laboratory stand at 20 °C.

**Figure 4 sensors-19-01558-f004:**
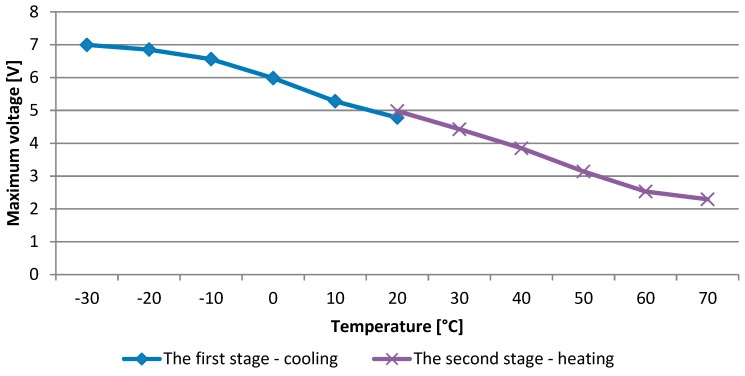
The dependence of the maximum voltage on the connected resistor as a function of the operating temperature of the tested system.

**Figure 5 sensors-19-01558-f005:**
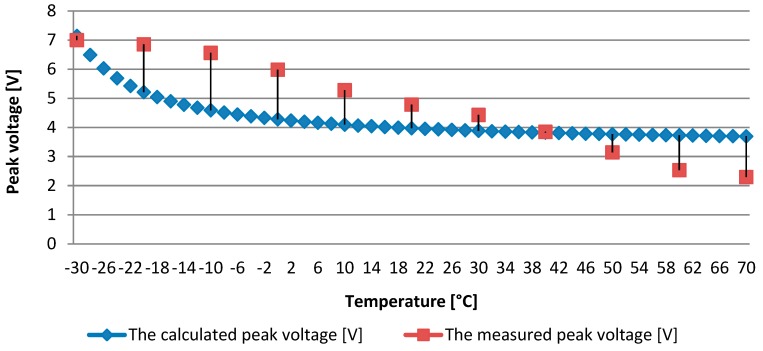
The calculated values of the peak voltage generated by the piezoelectric transducer juxtaposed with the measured values for different temperatures of operation.

**Table 1 sensors-19-01558-t001:** The geometric and material parameters of the tested transducer [1,24,27].

No.	Description	Notation (units)	Value
1	Active length of the MFC	$l_{MFC}\;\left(mm\right)$	85
2	Active length of the tested specimen	$l_{P}\;\left(mm\right)$	50
3	Active width of the MFC	$b_{P}\;\left(mm\right)$	14
4	Thickness of the MFC transducer	$h_{MFC}\;\left(mm\right)$	300
5	PZT fibres thickness	$h_{P}\;\left(\mu m\right)$	127
6	Capacitance	$C_{P}\;\left(nF\right)$	84.04
7	Copper fibre volume fraction	$V_{E}\;\left(dimensionless\right)$	0.19
8	PZT fibre volume fraction	$V_{P}\;\left(dimensionless\right)$	0.824
9	Piezoelectric constant	$d_{31}\;\left[\frac{pC}{N}\right]$	−1.7 × 10^+02^
10	Elastic compliance constant	$s_{11}^{E}\;\left[\frac{m^{2}}{N}\right]$	16.4 × 10^−12^
11	Dielectric constant	$\varepsilon_{33}^{T}\;\left[\frac{F}{m}\right]$	1.504 × 10^−08^

**Table 2 sensors-19-01558-t002:** The operating parameters of the tested Macro Fibre Composites (MFC) transducer.

Description	Notation (units)	Value
Resistance of the applied resistor	$R_{Z}\;\left(k\Omega \right)$	94
Frequency of excitation	f (Hz)	1.5
Maximum operational tensile strain	$S_{1}\;\left(ppm\right)$	2000

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
