# Peer review of "Modelling and Laboratory Tests of the Temperature Influence on the Efficiency of the Energy Harvesting System Based on MFC Piezoelectric Transducers"

_sensors, 2019, doi:10.3390/s19071558_

Reviewer 1 Report

This paper studies the temperature influence on efficiency of MFC piezoelectric transducer. A wide range of temperature is theoretically and experimentally studied. The content is interesting and it is recommended for publication after minor revision. There are some issues which need to be addressed:

1.      The introduction part needs to be improved. Some sentences need to be rewritten. Some general information related to piezoelectric materials can be removed and some latest studies in this field can be added to enhance the introduction section such as:

Zhao, L., & Yang, Y. (2018). An impact-based broadband aeroelastic energy harvester for concurrent wind and base vibration energy harvesting. Applied energy, 212, 233-243.

Izadgoshasb I, Lim YY, Lake N, Tang L, Padilla RV, Kashiwao T. Optimizing orientation of piezoelectric cantilever beam for harvesting energy from human walking. Energy conversion and management. 2018 Apr 1;161:66-73.  

Izadgoshasb I, Lim YY, Tang L, Padilla RV, Tang ZS, Sedighi M. Improving efficiency of piezoelectric based energy harvesting from human motions using double pendulum system. Energy Conversion and Management. 2019 Mar 15;184:559-70.

2.       Page 2 lines 86 and 89: references should be added.

3.       Line 88 to 91: Not clear, more explanation is needed.

4.       Page 3, line 132: “in the paper...” So why the experiment on MFC was conducted on fixed frequency?

5.       Page 5, line 204: The title of the paper can be removed and more explanation should be added.

6.       Page 6 line 258: Are all equations adopted from [25]? Clarification is needed.

7.       Page 10: Some paragraphs related to methodology are included in results section (such as line 346-360). Suggest to move them to methodology section.

8.       Page 12: Is the effect of using different epoxies and substrate materials (such as the MFC attached on the aluminium beam using stronger epoxy) during changing temperature considered in this research? Please elaborate.

9.       Writing of conclusion needs to be improved.

Author Response

Response to the Reviewers’ comments

Authors would like to thank the Reviewers for their work and extremely insightful and substantive opinion regarding the submitted paper as well as for some remarks that were very helpful for establish directions of further works.

The paper was corrected according all suggestions and all changes are highlighted using the "Track Changes" function.

The paper was also verified and corrected for the linguistic and grammatical mistakes in English by a professional translator. Changes introduced by the text verifier were marked only on the first page of the article using the "Track Changes" function due to their quite large number, which could blacken the transparency of the introduced corrections.

 Details of introduced corrections are presented in attached file and directly in the paper.

Kind regards,

Authors

Reviewer 2 Report

The paper reports the results of modelling and laboratory tests considering the temperature influence on the efficiency of the process of energy harvesting in a system, which makes use of an MFC piezoelectric transducer. The authors also propose a mathematical model, describing the tested phenomenon, which actually introduces the well-known electrical relationships.

The idea of powering electronic systems using piezoelectric components is well known and widely applied. However, the influences of environmental conditions on the systems’ efficiency is not yet deeply investigated, which constitutes interesting research case proposed by the authors.

Literature overview is well prepared. The proposed research extends the already known scope of investigations of the temperature influence, as regarding electromechanical impedance, SHM, PZT, numerical results, coupling coefficients, PZT for energy harvesting. The authors specifically focus their attention on MFC used for energy harvesting. The soundness and novelty level of the paper are considered as limited however there is a gradual gain of knowledge observed, which is worth to be published after the following comments/corrections are considered in the manuscript.

In eq. (1) and (2) and others there are no components introducing the modeled temperature influence. There is no connection between physics (influence of temperature) and the model used to simulate energy harvesting.

There are some language flaws and the manuscript needs to be corrected thoroughly, mainly in terms of the use of the articles “-”, “a/an” and “the”.

The expressions that deed improvement:

Row 17: “phenomenon” -> “phenomena”

Row 19: “model was verified by laboratory tests” -> “model was validated by laboratory tests”

Row 21: “extortion” -> “excitation”

Row 44: “no necessary to” -> “no necessity to”

Row 64: “properly bent system” what does it mean?

Row 91-92 -  please remove the title of the cited paper

Row 204-205 – the same comment

Author Response

(The authors gave the same response as above.)
